# Quantum Chaos and Level Dynamics

**DOI:** 10.3390/e25030491

**Published:** 2023-03-13

**Authors:** Jakub Zakrzewski

**Affiliations:** 1Institute of Theoretical Physics, Faculty of Physics, Astronomy and Applied Computer Science, Jagiellonian University, Łojasiewicza 11, 30-348 Kraków, Poland; jakub.zakrzewski@uj.edu.pl; 2Mark Kac Complex Systems Research Center, Jagiellonian University, 30-348 Kraków, Poland

**Keywords:** 80th birthday of Giulio Casati, quantum chaos, level dynamics

## Abstract

We review the application of level dynamics to spectra of quantally chaotic systems. We show that the statistical mechanics approach gives us predictions about level statistics intermediate between integrable and chaotic dynamics. Then we discuss in detail different statistical measures involving level dynamics, such as level avoided-crossing distributions, level slope distributions, or level curvature distributions. We show both the aspects of universality in these distributions and their limitations. We concentrate in some detail on measures imported from the quantum information approach such as the fidelity susceptibility, and more generally, geometric tensor matrix elements. The possible open problems are suggested.

## 1. Introduction

It is a great pleasure to be able to contribute to the volume of Giulio Casati’s 80th birthday. Since the very beginning of my encounter with quantum chaos, Giulio Casati was one of those whose works inspired younger people. As an example, let me mention a contribution of late Prot Pakoński with whom I had a pleasure to consider the Kepler map (one of the toy models of Giulio), extending it to arbitrary polarization of the microwaves [1]. In this review I will discuss, however, a different topic—the statistical measures related to level dynamics in quantally chaotic systems. This is where we met scientifically, contributing to a single paper I had an honour to coauthor with Giulio [2]. The subject of level dynamics developed mainly in the 1980s and 1990s of the former millennium, yet it has recently found interesting extensions and applications in the modern many-body physics.

Level dynamics, described in a pedagogical way in the books of late Fritz Haake [3] and Hans-Jürgen Stöckmann [4], considers the motion of levels as a function of some arbitrary scalar parameter λ which characterizes the Hamiltonian H(λ) of the system. It may be viewed as the motion of interacting fictitious particles (represented by levels) with λ being the effective time, as described originally by Pechukas [5] and followed by Yukawa [6] who built a corresponding statistical mechanics picture. This was applied in different ways to either justify the random matrix theory application to quantum chaotic spectra (this aim has been, in fact, accomplished by a beautiful periodic orbit theory application by the late Fritz Haake and collaborators—for a review see the Ref. [3]) or to define new statistical measures of energy spectra and find their distributions. Without any claim for completeness, some of these applications will be reviewed below.

We begin with defining the notation for level dynamics and the corresponding statistical mechanics picture in Section 2 showing how standard results from this approach provide a prediction for level statistics in a chaotic integrable transition. The resulting distribution seems to work surprisingly well for the data in many-body localization crossovers [7]. To somehow complete the picture, we review in the next Section several other models for the statistics in the transition, notably the banded matrix model, developed by Casati and coworkers [8,9,10,11,12]. In Section 4 we review the universality conjecture in level dynamics [13] showing how it is reflected in the so-called curvature distributions [14] in Section 5. We mention the velocity correlations in Section 6 and the avoided crossings statistics [15,16] in Section 7. Applications of these measures are discussed, stressing their limitations in real systems. Section 8 describes, on the other hand, very recent findings on the distribution of fidelity susceptibility [17], while the extensions to many parameter dynamics with geometric tensor matrix elements distributions are reviewed in Section 9. We mention briefly the parametric measures in the transition to localized regimes in Section 10. We finish with conclusions discussing future perspectives.

While the presentation that follows is only theoretical, we should mention that the predictions concerning level dynamics were tested, to some extent, in experiments. As typical for quantum chaos, those experiments were not carried out on eigenvalues of the Schrödinger equation, but rather on related models of quasi-2D microwave cavities or propagation of acoustic waves. We provide an incomplete list of references to beautiful experiments [18,19,20,21,22,23,24] stressing the work of the Stöckmann group [18] where a rather complete comparison of different measures with experimental microwave resonance data was carried out.

## 2. Level Dynamics

Let us recall some basic details on level dynamics, to fix the notation. Let the Hamiltonian, $H\left(\lambda \right)=H_{0}+\lambda V$, depend on some parameter λ for arbitrary $H_{0}$ and *V*. The eigenvalue equation
(1)$$\begin{array}{c} H\left(\lambda \right)|a\left(\lambda \right)\rangle =E{\left(\lambda \right)}_{a}|a\left(\lambda \right)\rangle , \end{array}$$(where $E{\left(\lambda \right)}_{a}$ is the eigenvalue corresponding to eigenvector |a(λ)〉) upon differentiation with respect to λ gives
(2)$$\begin{array}{c} \frac{d}{d\lambda }E_{a}\equiv {\dot{E}}_{a}=\langle a|V|a\rangle \equiv V_{aa}. \end{array}$$Let us call $p_{a}\equiv {\dot{E}}_{a}$ a velocity of level $E_{a}$ where λ becomes a fictitious time. Such a “motion” for a set of levels $\left\{E_{a}\left(\lambda \right)\right\}$ is visualized in Figure 1. A λ derivative (denoted by a dot over the variable) of $p_{a}$ yields
(3)$$\begin{array}{c} {\dot{p}}_{a}=2\sum_{b\neq a}\frac{V_{ab}V_{ba}}{E_{a}-E_{b}}=2\sum_{b\neq a}\frac{|f_{ab}{|}^{2}}{{\left(E_{a}-E_{b}\right)}^{3}}, \end{array}$$
where we introduced $f_{ab}=V_{ab}\left(E_{a}-E_{b}\right)$. Following the procedure one step further, we find equations for $\dot{f_{ab}}$:(4)$${\dot{f}}_{ab}=\sum_{r\neq a,b}^{}f_{ar}f_{rb}\left[\frac{1}{{\left(E_{a}-E_{r}\right)}^{2}}-\frac{1}{{\left(E_{b}-E_{r}\right)}^{2}}\right].$$One notices that no new quantities appear, and the set of Equations (Equation 2)–(Equation 4) is closed. It is sometimes called the Pechukas–Yukawa equations following original works [5,6]. This set of nonlinear equations is integrable—this is not surprising as a problem is equivalent to a diagonalization of the Hamiltonian matrix as noted in the Ref. [3].

The eigenvalues of $H=H_{0}+\lambda V$ are *V*-dominated for large λ and the dynamics become trivial. Haake [3] introduces a different λ dependence (equivalent for small λ): $H\left(\lambda \right)=\sqrt{f}\left(H_{0}+\lambda V\right)$ with $f={\left(1+\lambda^{2}\right)}^{-1}$ while we shall follow the “trigonometric choice” [4,14,16]
(5)$$H=H_{0}\cos\left(\lambda \right)+V\sin\left(\lambda \right).$$This results in an additional harmonic binding of eigenvalues which prevents them from escaping to infinity. In effect, the equations of motion become:(6)$$\begin{array}{c} {\dot{E}}_{a}=\langle a|\dot{H}|a\rangle =p_{a} \end{array}$$
and
(7)$$\begin{array}{c} {\dot{p}}_{a}=-E_{a}+2\sum_{b\neq a}\frac{|f_{ab}{|}^{2}}{{\left(E_{a}-E_{b}\right)}^{3}} \end{array}$$
with $f_{ab}=\langle a|\dot{H}|b\rangle \left(E_{a}-E_{b}\right)$.

Since the dynamics are integrable, the appropriate statistical description should involve all possible constants of the motion. Such an approach would be a formidable task. The Yukawa simplified way is just to consider the simplest integrals of motion, the total energy *H* and the trace of the square of $f_{ab}$ matrix, $Q=\frac{1}{2}\mathrm{Tr}\left(F^{2}\right)$ [4] with
(8)$$H=\frac{1}{2}\sum_{n}^{}\left(p_{n}^{2}+E_{n}^{2}\right)+\frac{1}{2}\sum_{n,m}^{}\frac{|f_{nm}{|}^{2}}{{\left(E_{n}-E_{m}\right)}^{2}}.$$The phase space density, according to Gibbs, is:(9)$$\rho =\frac{1}{Z}\exp\left(-\alpha H-\gamma Q\right).$$The reader may be surprised that we use α as an effective inverse temperature. The simple reason is that, in order to follow the sacred quantum chaos notation, we reserve β for a level repulsion parameter with β=1,2,4 characterizing different universality classes of Dyson and corresponding, for Gaussian ensembles to Gaussian Orthogonal, Unitary, and Symplectic Ensembles (denoted as GOE, GUE, and GSE, respectively). The density ρ may be explicitly written out as
(10)$$\rho =\frac{1}{Z}\exp\left(-\alpha \left[\frac{1}{2}\sum_{n}^{}\left(p_{n}^{2}+E_{n}^{2}\right)+\frac{1}{2}\sum_{n,m}^{}\frac{|f_{nm}{|}^{2}}{{\left(E_{n}-E_{m}\right)}^{2}}\right]-\gamma \frac{1}{2}\sum_{n,m}^{}|f_{nm}{|}^{2}\right).$$By integrating out the variables $p_{n}$ and $f_{nm}$, we can compute the joint probability distribution (JPD) of eigenvalues [4]
(11)$$P\left(E_{1},E_{2},\ldots \ldots ,E_{n}\right)∼\prod_{n<m}^{}|\frac{{\left(E_{n}-E_{m}\right)}^{2}}{1+\frac{\gamma }{\alpha }{\left(E_{m}-E_{n}\right)}^{2}}{|}^{\beta /2}\exp\left(-\frac{\alpha }{2}\sum_{n}^{}E_{n}^{2}\right),$$
with β=1,2,4 corresponding to three universality classes. The β appears in (Equation 11) as the structure of the integrated variables $f_{nm}$ depends on the universality class with *F* being orthogonal, unitary or symplectic. The similar ensemble was considered by Gaud in the Ref. [25], as well as Forrester [26] and Hasegawa and Ma [27]. They considered mainly two point correlation functions for the unitary ensemble. We rather concentrate on the time-reversal invariant case, as most commonly met in current many-body localization studies.

Equation (Equation 11) is simplified in different limiting cases. The Poissonian distribution was reached in the γ/α>>1 limit. The distribution becomes
(12)$$P\left(E_{1},E_{2},\ldots \ldots ,E_{n}\right)∼\exp(-\frac{\alpha }{2}\sum_{n}^{}E_{n}^{2}).$$On the other hand, for γ/α≪1 the distribution yields the Gaussian ensemble result. We have
(13)$$P\left(E_{1},E_{2},\ldots \ldots ,E_{n}\right)∼\prod_{n>m}^{}|E_{n}-E_{m}{|}^{\beta }\exp(-\frac{\alpha }{2}\sum_{n}^{}E_{n}^{2}).$$Finally, to reach (GOE) in this limit, we fix β=1, and we also fix, for simplicity, α=1 (the latter choice affects the global energy scale only). We also denote $\gamma /\alpha =10^{p}$. The distribution (Equation 11) takes the final form
(14)$$P\left(E_{1},E_{2},\ldots \ldots ,E_{n}\right)∼\prod_{n<m}^{}|\frac{{\left(E_{n}-E_{m}\right)}^{2}}{1+10^{p}{\left(E_{m}-E_{n}\right)}^{2}}{|}^{1/2}\exp\left(-\frac{1}{2}\sum_{n}^{}E_{n}^{2}\right),$$
where $p=\log_{10}\frac{\gamma }{\beta }$ is the single parameter interpolating between GOE (p→−∞) and Poisson (p→∞) limits. The first term in (Equation 14) represents the pairwise interaction between the particles and the exponential term provides the harmonic binding of the eigenvalues. The resulting distribution, obtained using Monte-Carlo sampling for different *p*, was shown to faithfully reproduce statistics of eigenvalues on the transition between ergodic and many body localized situations [7].

## 3. Other Interpolating Ensembles

It is interesting to review several interpolating statistics models proposed in the past. An early work of Rosenzweig and Porter [28] is certainly worth mentioning. In this model the variance of off-diagonal elements in random matrices is controlled by a matrix dimension-dependent parameter. Its value interpolates between the Gaussian orthogonal ensemble (GOE) value to vanishing values for the Poissonian case. The other approach was proposed on the basis of a Wigner-inspired 2×2 matrix approach by Lenz and Haake [29]. Yet another was an ad hoc expression known as Brody distribution [30] which fit to low-resolution experimental data surprisingly well. Berry and Robnik [31] proposed the distribution based on sound physical assumption of the separation between “chaotic” wave functions faithful to GOE and those localized in the regular part of the phase space. The corresponding distribution was shown to work well in the so-called deep semiclassical limit [32]. Another proposition due to Seligman and coworkers [33] assumed that the variance of off-diagonal elements $a_{ij}$ should scale as $\exp[-{\left(i-j\right)}^{2}/\sigma^{2}]$. For σ→0 one recovers the Poisson case while σ→∞ becomes GOE. Yet another well-known approach is that of Guhr [34] who used supersymmetric techniques to express the two-level correlation function in the Poisson-GOE ensemble in terms of a double integral. It is worth stressing that another popular proposition was advocated by Giulio Casati and coworkers [8,9]. They considered banded Gaussian random matrices as a useful tool in describing the transition, where the corresponding parameter was $y=b^{2}/N$ with *b* being the matrix bandwidth and *N* its rank.

While the (unfolded) level spacing statistics was the main object of quantum chaos studies, in a many-body localization (MBL) context Huse and Oganesyan [35] introduced a new measure, the gap ratio, defined as $r_{n}=min\left[\delta_{n},\delta_{n-1}\right]/max\left[\delta_{n},\delta_{n-1}\right]$, where $\delta_{n}=E_{n+1}-E_{n}$ is the energy gap between the consecutive energy levels. The dimensionless gap ratio does not require unfolding. The MBL transition description was addressed by Serbyn and Moore [36] who proposed two stages of the GOE–Poisson transition: (1) A Short Range Plasma Model (SRPM) and (2) a semi-Poissonian level statistics [37,38]. Recent efforts worth mentioning are a β-Gaussian (β−G) model [39]. A comparison of the performance of different models was given in the Ref. [40], while the Ref. [41] proposes a more complicated, two-parameter β−h model, where the pairwise interaction between the levels is limited to a number *h*.

Comparison of some of these models with numerics for interacting disordered spin systems modelling ergodic to the MBL transition is given in the Ref. [7]. The interested reader should consult the Ref. [7] for details, but it suffices to say here that the single-parameter Yukawa-like model described above compared favorably with other single-parameter models and quite faithfully reproduced the disordered spin data for the MBL–ergodic crossover.

## 4. Universality of Parametric Dynamics

A simple inspection of Equation (Equation 10) shows that the velocities, $p_{n}$, have, in this approach, a Gaussian distribution with the variance determined by the “inverse temperature” α. This is the essence of level dynamics universality, as was determined and thoroughly studied by Simons and Altschuler [13,42]. The level spacings have a single scale—the mean level spacing, Δ. The unfolding then corresponds to rescaling the energy levels $\epsilon_{i}=E_{i}/\Delta$. Level dynamics introduces a novel scale determining how fast the eigenvalues change with the parameter λ. The original definition [13] involves the velocity–velocity correlation function for unfolded levels
(15)$$C\left(\lambda \right)=\langle p_{n}\left(0\right)p_{n}\left(\lambda \right)\rangle /\Delta^{2}$$
averaged over eigenstates *n*. Then C(0) yields the second, apart from the mean level spacing, important scale. When the levels are unfolded using mean spacing, Δ, and the parametric dependence is “unfolded” using C(0) as [13]
(16)$$x=\sqrt{C(0)}\lambda$$
the spectral properties of different systems should be similar. Clearly, C(0) in our notation is directly related, modulo-unfolding to the “inverse temperature” α in the Gibbs ensemble.

There is a hidden assumption in the level dynamics universality as formulated by Simons and Altschuler [13,42] and apparent in the Pechukas–Yukava statistical approach—the parameter change is global in a sense that both $H_{0}$ and *V* are of similar “strength” i.e., belong to the same ensemble. This does not have to be so, as nicely exemplified in the Ref. [43] where it is shown that small local perturbation modifies strongly the velocity distribution in the billiard example studied. Assuming random wavefunctions it is derived that in such a case $P\left(v\right)\propto K_{0}\left(A|v|\right)$ where *A* is a constant and $K_{0}\left(.\right)$ denotes a modified Bessel function. The distinction between local and global perturbations in the context of unviersality were further studied in the Ref. [44].

It seems natural to review now the properties of velocity correlation function. For the reasons that become obvious later, it is more convenient to consider first the second derivatives of energy levels with respect to the parameter, the so-called curvatures.

## 5. Curvature Distributions

The curvatures of levels $K_{n}={\dot{p}}_{n}$ as derivatives of velocities should be called in the dynamics language “level accelerations”. We stick to curvatures as this is a commonly used name. Large curvatures appear in the vicinities of avoided crossings in the system. Then, essentially only two levels are involved. Following this strategy [45] showed that the large curvature tail behaves as ${\left|K\right|}^{-(\beta +2)}$ for all three universality classes.

The full analysis of curvature distributions, not limited to large curvature tail, was carried out in the Ref. [14]. Large numerical data collected for all three ensembles suggested the following simple and analytic form:(17)$$P\left(K\right)=N_{\beta }\frac{1}{{\left(1+{\left(K/\gamma_{\beta }\right)}^{2}\right)}^{\frac{\beta +2}{2}}}$$(with β=1,2,4 for GOE, GUE and GSE, respectively) and
(18)$$\gamma_{\beta }=\pi \beta C(0)\Delta$$
where, recall, Δ is the mean level spacing (i.e., an inverse of the mean density of states). Defining the dimensionless curvature, *k*, as
(19)$$k=\frac{K}{\gamma_{\beta }}$$
we have explicitly
(20)$$\begin{array}{ccc} P_{O}\left(K\right) & = & \frac{1}{2}\frac{1}{{\left(1+k^{2}\right)}^{3/2}} \end{array}$$
(21)$$\begin{array}{ccc} P_{U}\left(K\right) & = & \frac{2}{\pi }\frac{1}{{\left(1+k^{2}\right)}^{2}} \end{array}$$
(22)$$\begin{array}{ccc} P_{S}\left(K\right) & = & \frac{8}{3\pi }\frac{1}{{\left(1+k^{2}\right)}^{3}}. \end{array}$$These expressions, which could be claimed as being determined via Monte-Carlo integration and inspired guess, were soon proven analytically for all three ensembles of Gaussian random matrices [46,47] and by an alternative technique in the Refs. [48,49].

Let us remark that the above definition of *k* differs from the form suggested by the universality rule, (Equation 16), $d^{2}\epsilon /dx^{2}$ by a multiplicative factor πβ which simplifies (Equation 20)–(22).

The distributions (Equation 20)–(22) appear to work well for circular ensembles as well as some quantally chaotic systems such as kicked tops [14] or periodic band random matrices in the metalic regime, as shown by Casati and coworkers [11]. The question remains, however, to what extent these RMT predictions are universal and to what extend the particular quantally chaotic systems are faithful to them. The first aspect was clarified by Li and Robnik [50] who pointed out that a nonlinear transformation from λ to some other parameter μ(λ) leads to a different curvature distribution as the transformation between curvatures is nonlinear. It reads [50]:(23)$$k_{\mu }=k_{\lambda }-\frac{p_{\lambda }}{\pi \beta \langle p_{\lambda }^{2}\rangle }\frac{\mu^{\prime \prime }}{\mu^{\prime }}.$$In the expressions above $k_{\mu }$ and $k_{\lambda }$ are normalized curvatures calculated with respect to the corresponding parameters, $p_{\lambda }$-the slope and prime denotes derivative with respect to λ. As Li and Robnik [50] point out since velocities are Gaussian distributed (fast decaying) the universality of curvatures may be restored for large curvatures but the nonlinearity of the transformation (Equation 23) prevents universality at all scales, see also [51]. The same argument shows, however, that for any “local” linear transformation the universality may hold. As long as changes of H(λ) are linear in λ, as assumed in the derivation above, one might expect the universality to hold.

There is, however, another origin for the lack of universality which gives us insight into the physics involved. Already Takami and Hasegawa [52] suggested that the presence of scarring, i.e., strong localization of eigenstates in the space where unstable periodic orbits exist in the classical limit [53,54] may affect curvatures. Numerical studies of several examples such as the hydrogen atom in a magnetic field or quantum billiard proved that this is indeed the case [14]. While referring the reader to an original paper for numerical details it suffices to say that strong scarring leads to a peculiar level dynamics with some levels (scarred eigenstates) have quite different slope than the rest and interact with other levels only locally in narrow avoided crossings. Those levels behave like solitons and may be described as such [55,56]. Their behavior leads to an excess of small curvatures (outside of these avoided crossings) as well as very large curvatures (at the centers of avoided crossings).

It is worth stressing (which we just do with a single sentence) that the curvatures are strongly linked with transport and conductance [11,57]. Particularly interesting for this case are situations where the parameter breaks time reversal invariance as it happens for twisted boundary conditions.

## 6. Velocity Correlations

Let us come back to the level slopes, i.e., velocity correlations. Simons and Altschuler [13,42] in their analysis considered the autocorrelator of velocities at a some energy difference, ω, $\tilde{c}\left(x,\omega \right)$ (note—this is a different object than C(x) which involves correlations for the same level *n*) that involves all level velocities in a given interval studied. We refer the reader to original papers for details [13,42]. The C(λ) or rather C(x) was studied numerically [58] for all three unitary classes. A simple analytic approximation for C(x) was proposed in terms of the plasma error function, see [58].

The large *x* limits was elegantly solved [13,42] yielding $C\left(x\right)=-2/\beta \pi^{2}x^{2}$ for GUE. Interesting information may be obtained from a small *x* limit when $\tilde{c}\left(x,\omega \right)$ (for ω=0) and C(x) behave similarly.

Application of Taylor series expansion of C(x) allows one to link the velocity correlator to the variance of the rescales curvatures. Explicitly, one obtains [58]
(24)$$C\left(x\right)=C\left(0\right)(1-\frac{1}{2}\beta^{2}\pi^{2}x^{2}\langle k^{2}\rangle ).$$Defining normalized correlation c(x)=C(x)/C(0) one reproduces the results [42] for GUE: $c_{\mathrm{GUE}}\left(x\right)=1-2\pi^{2}x^{2}$ and gets $c_{\mathrm{GSE}}\left(x\right)=1-\frac{8}{3}\pi^{2}x^{2}+\ldots$ for the symplectic ensemble [58]. Interestingly, for the most common orthogonal universality class one encounters the problem as the variance of curvatures, following (Equation 20), does not exist. This indicates that the small *x* behavior may be singular and the Taylor expansion is questionable.

This issue has been studied further in the Ref. [2] where it was shown that in fact c(x) reveals singularities around x=0. Taking the parametric dependence (Equation 5) one may show that Fourier components of the Fourier transform of c(x) have algebraic tails which directly indicates singularities at x=0 of the velocity correlator. Again we just quote the the final result which shows that
(25)$$\begin{array}{c} c_{\mathrm{GOE}}\left(x\right)∼1+b_{1}x^{2}\left|\ln\left(x\right)\right|,\\ c_{\mathrm{GUE}}\left(x\right)∼1-2\pi^{2}x^{2}+b_{2}\left|x^{3}\right|,\\ c_{\mathrm{GSE}}\left(x\right)∼1-\frac{8}{3}\pi^{2}x^{2}+b_{3}x^{4}+b_{5}\left|x^{5}\right|, \end{array}$$
with $b_{i}$ being coefficients of the order of unity. One may observe that the singularity at x=0 becomes weaker with growing level repulsion β, being most severe for GOE. An even more in depth analysis of singularities appears in the Ref. [59] where explicit values for the parameters, $b_{i}$ are found.

As mentioned in Section 4 local (instead of global) perturbation affect strongly velocity distributions [43,44]. This has also a pronounced effect on velocity correlations [43,44] as further analysed in detail for the unitary ensemble [60].

## 7. Avoided Crossings Distributions

Another statistical property with interesting links to level spacings is the distribution of avoided crossings, i.e., minima of distances between neighboring levels. The problem of finding the corresponding distribution was formulated by Wilkinson [61] who has shown that integrated distributions for small minimal distances *D* for GOE (GUE) show similar repulsion as present in spacing distributions. Avoided crossings for billiard models were numerically studied by Goldberg and Schweizer [62]. While in a general case the exact distributions are not known in some analytic form, a well working approximations based on two-levels approximation may be easily derived following the Wigner approach for level spacings themselves [15]. For GOE case it is written down immediately as the two-level Hamiltonian $H=H_{0}+\lambda V$ may be written in the eigenbasis of *V* as
(26)$$H=\left[\begin{array}{cc} a & d\\ d & b \end{array}\right]+\lambda \left[\begin{array}{cc} v_{1} & 0\\ 0 & v_{2} \end{array}\right].$$The minimal distance between levels is simply 2|d|. Since $H_{0}$ is assumed to correspond to GOE, *d* is Gaussian distributed, so we get the distribution (for D=2|d|) normalized to unit mean avoided crossing:(27)$$P\left(D\right)=\frac{2}{\pi }\exp\left[-\frac{D^{2}}{\pi }\right],\;D>0.$$Situation is only slightly more complicated for other ensembles. For GUE *d* in (Equation 26) should be complex, $d=d_{1}+id_{2}$, with independently Gaussian distributed (with the same variance) $d_{i}$. A simple integral leads to a normalized distribution
(28)$$P\left(D\right)=\frac{\pi D}{2}\exp\left[-\frac{\pi }{4}D^{2}\right],$$
which is identical to the so-called Wigner surmise for spacings for GOE. We observe a simple rule that the avoided crossings in two-level approximation share the same distribution as the nearest neighbor spacings but the the repulsion parameter β reduced by unity. So for GUE with β=2 we get the Wigner formula corresponding to spacings for β=1. This is in full agreement with small *D* perturbative prediction of the Ref. [61]. Similarly, an explicit calculus shows that for β=4 GSE the avoided crossing distribution behaves as $D^{3}$ for small *D*.

Numerical tests (which have to be carefully done to correctly estimate and avoided crossing values [16]) show excellent agreement between two-level approximate formulae and numerical data for random matrices of larger sizes. The agreement is in fact better than the spacings and the Wigner surmise. The reason is simple—the two-level approximation works better for minimal distances between levels.

## 8. Fidelity Susceptibility

Rapidly developing in the past 20 years, the quantum information field brought yet another measure which may be related to level dynamics, the fidelity, F [63]. While generally defined for mixed states, a pure-state definition [64] suffices for our purposes
(29)F=|〈ψ(0)|ψ(λ)〉|.Here, ψ(λ) is an eigenstate at the value of the parameter equal to λ. Taylor expansion for small λ leads to the definition of fidelity susceptibility, χ
(30)$$F=1-\frac{1}{2}\chi \lambda^{2}+O\left(\lambda^{3}\right).$$Fidelity susceptibility became an indicator of quantum phase transitions. At the transition point, the ground state properties change, leading to the enhancement of χ [64,65,66]. Apart from ground states, thermal states were also considered [67,68,69]. The fidelity statistics, taking into account a set of eigenstates, was discussed for the first time in the Ref. [17]; note that an attempt to identify many-body localization transition was due to the Ref. [70]. We shall briefly review the results of the Ref. [17] that provide one of the rare situations when exact analytic results are available for the arbitrary size of random matrices.

Consider $H=H_{0}+\lambda V$ with both $H_{0}$, *V* belonging to GOE or GUE. Fidelity susceptibility of state |n〉 with energy $E_{n}$ of $H_{0}$ is easily derived as
(31)$$\chi_{n}=\sum_{m\neq n}\frac{|V_{nm}{|}^{2}}{{\left(E_{n}-E_{m}\right)}^{2}},$$
showing some similarity to curvatures ${\dot{p}}_{n}$ (Equation 3)—the difference is just a power in the denominator.

The probability distribution of the fidelity susceptibility at energy *E* reads:(32)$$P\left(\chi ,E\right)=\frac{1}{N\rho (E)}\langle \sum_{n=1}^{N}\delta \left(\chi -\chi_{n}\right)\delta \left(E-E_{n}\right),\rangle$$
which we consider at the center of the spectrum E=0. Following the technique developed in the Refs. [48,71,72], one arrives [17] at, for the GOE case,
$$P_{N}^{O}\left(\chi \right)=\frac{C_{N}^{O}}{\sqrt{\chi }}{\left(\frac{\chi }{1+\chi }\right)}^{\frac{N-2}{2}}{\left(\frac{1}{1+2\chi }\right)}^{\frac{1}{2}}\left[\frac{1}{1+2\chi }+\frac{1}{2}{\left(\frac{1}{1+\chi }\right)}^{2}I_{N-2}^{O}\right],$$
where $C_{N}^{O}$ is a normalization constant and
(33)$$I_{N}^{O}=\left\{\begin{array}{c} N\frac{N+2}{N+3/2},\;\;N\;\mathrm{even},\\ N+1/2,\;\;N\;\mathrm{odd}. \end{array}\right.$$Equation (33) is an exact result for an arbitrary rank *N* of the random matrix from GOE. This is one of the rare situations when analytic formulae for arbitrary *N* and not only for the N=2 or N→∞ limit are known.

The N→∞ limit is interesting. As $I_{N}^{O}$ is asymptotically proportional to *N*, one can define a scaled fidelity susceptibility y=χ/N. Its distribution, in the N→∞ limit takes a rather simple form
(34)$$P^{O}\left(y\right)=\frac{1}{6}\frac{1}{y^{2}}\left(1+\frac{1}{y}\right)\exp\left(-\frac{1}{2y}\right).$$As tested numerically, this expression works well for N∼200 already.

Similarly, see the Ref. [17] for the derivation, where one obtains an analytic, valid for arbitrary *N* results, for GUE. We quote here just the N→∞ limit for the scaled fidelity susceptibility
(35)$$P^{U}\left(y\right)=\frac{1}{3\sqrt{\pi }}\frac{1}{y^{5/2}}\left(\frac{3}{4}+\frac{1}{y}+\frac{1}{y^{2}}\right)\exp\left(-\frac{1}{y}\right),$$
while for a full expression for χ valid for arbitrary *N* as well as for a comparison with numerical data, we refer to the Ref. [17].

The distributions (33) and (34) for GOE and GUE, respectively, are presented in Figure 2. For large scaled fidelities *y* a power law decay is observed $1/y^{2}$ for GOE and $1/y^{5/2}$ for GUE.

## 9. Generalization to More Parameters

A natural extension of parametric level dynamics occurs in the presence of more than one parameter. One may define
(36)$$H=H_{0}+\sum_{i}\lambda_{i}V_{i}$$
where $H_{0}$, $V_{i}$ are statistically independent and drawn from the same (as in this review) or different universality classes. Obviously, novel problems appear in that case which we describe briefly only. Probably it was Michael Wilkinson and his student, Elisabeth Austin, who addressed first such a situation in their study of density of degeneracies, for example, diabolical points [73]. A three-parameter family was considered for Chern integer fluctuations [59,74]. Steuwer and Simons [75] found the distribution of adiabatic curvature (related to Berry phase) for GUE. The multiparameter dynamics was recently revisited by Berry and Shukla who discussed Berry curvature deriving two-level and three-level distributions [76,77,78]. They found the large curvature scaling, $P\left(c\right)∼c^{-2}$ for the orthogonal and $P\left(c\right)∼c^{-5/2}$ for the unitary class, the result already present in the Ref. [75].

Importantly, however, Berry and Shukla [78] linked the problem with the quantum information concept of geometric tensor and the distance between quantum states [79,80,81]. The Fubini–Study distance in the Hilbert space between states differing by a small change of parameters form $\vec{\lambda }\equiv \left(\lambda_{1},\lambda_{2},\ldots ,\lambda_{n}\right)$ to $\vec{\lambda }+d\vec{\lambda }$ is
(37)ds2=1−|〈n(λ→)|n(λ→+dλ→)〉|2=∑ijRegij(n)(λ→)dλidλj,
where $g_{ij}^{\left(n\right)}\left(\vec{\lambda }\right)$ is the so-called geometric tensor [79,80,81] which governs the quenches (in λ) of the system. For (Equation 36)
(38)$$g_{ij}^{\left(n\right)}=\sum_{m(\neq n)}\frac{\langle n|V_{i}\left|m\rangle \langle m\right|V_{j}|n\rangle }{{\left(E_{n}-E_{m}\right)}^{2}}.$$Note that the distance between states is determined by the real part of the geometric tensor only—(Equation 37). The imaginary part, Imgij(n), gives the Berry curvature [78,82] related to changes of two parameters $\lambda_{i},\lambda_{j}$. For a single-parameter problem the geometric tensor reduces to a scalar proportional to the fidelity susceptibility discussed in the previous Section. Additionally, if $V_{i}$ belongs to the same ensembles, then the distributions of $g_{ii}$ are equal. One may consider, therefore, the trace $G=\mathrm{Tr}g_{ij}^{\left(n\right)}$ as an equivalent of the fidelity susceptibility while the distribution of the imaginary part of the geometric tensor reduces to the Berry curvature distribution.

The distribution of trace, *G*, is, therefore, given by (33) discussed above, valid for arbitrary *N* [17] after taking into account the fact that, for equal diagonal elements the trace is simply the matrix rank multiplying the diagonal element. The alternative derivation using supersymmetric techniques is provided in the Ref. [82] for GUE in the N→∞ limit. The same authors obtained for the Berry curvature the result derived earlier by the Ref. [75].

## 10. Towards the Localization Limit

While in the introductory part we have considered the transition between an integrable (localized) regime in the context of level spacings and the dynamics of Pechukas–Yukawa gas, later we mainly described the results for the ergodic regime well-described by Gaussian random ensembles. Here we shall briefly mention some of the results for level dynamics measures that involve the transition.

Here the seminal contributions were provided by the Como group centered around Giulio Casati [8,9,10,11]. The banded random matrix ensemble provided a natural tool to study the transition from ergodic (metallic) to localized transition by varying the width of the band. The team addressed also curvature distributions [2,11]. The analytic approach to the problem was pursued by Yan Fyodorov, who, starting around 1994, considered comprehensively level dynamics features close to the localization transition studying velocity correlations [83,84] or curvature distributions [85]. In particular, the velocity distribution for one-dimensional disordered wire is derived using the supersymmetric approach to be
(39)$$P\left(v_{s}\right)=\frac{\pi }{\sinh^{2}\left(\pi v_{s}\right)}\left\{\pi v_{s}\coth\left(\pi v_{s}\right)-1\right\},$$
for the scale velocity $v_{s}$. The curvature distributions were more recently addressed in the context of MBL studies [86,87]. The level dynamics across the many-body localization transition for the paradigmatic XXZ spin model was considered in the Ref. [88]. Velocity, curvature and fidelity susceptibility distributions were considered. Interestingly, while velocities depended on the choice of the parameter (being the interaction strength or the kinetic tunneling), curvature exhibited universal behavior in the delocalized regime. In the localized regime, curvature distributions reveal system specific characteristics that exemplify the presence of local integrals of motion in the localized phase. Large curvature or large fidelity susceptibility tails change their slope when entering the localized regime. Such behavior is well-understood qualitatively and linked to weaker level repulsion. One may also mention here a proposition to use adiabatic eigenstate deformation as a probe of entering the localized regime [89].

Recent studies address avoided crossing distributions and level dynamics on the localized side of the MBL transition [90,91,92]. Such an analysis, however, has to face the problem of assumed integrability which leads necessarily to exact crossing of levels associated with different quantum numbers.

## 11. Conclusions—Where Do We Stand

With this traipse through the last 40 years, starting with the Pechukas model [5], we hope to have shown that many fascinating results have been obtained in the studies of level dynamics of complex systems, but that there are still many open questions and unsolved problems. There are at least two areas that await a more decisive attack and, hopefully, solutions. One is the transition between different ensembles [93]. We have not, on purpose, reviewed only a few studies in this domain for the reader to formulate their own problems. A second related area with several white spots lies in multiparameter level dynamics. The latter has been mostly limited to studies within unitary ensembles, notoriously easiest to treat. We do not know the Berry curvature distribution for the orthogonal ensemble. We do not know about the geometric tensor properties when different parameters induce different transitions. Even the simplest questions remain unanswered. For example, simple analysis shows that fidelity susceptibility of Berry curvature decays with reverse quadratic power for GOE while the corresponding power is −5/2 for GUE. Can we say something when we couple GOE and GUE ensembles by some parameter, such as weakly breaking the time reversal invariance? Can we generalize the findings to the symplectic ensemble? What about the ten-fold method [94]? It is my belief that we may expect in the future some very interesting results coming from new people entering the subject. I already anticipate the excitement.

## Figures and Tables

**Figure 1 entropy-25-00491-f001:**
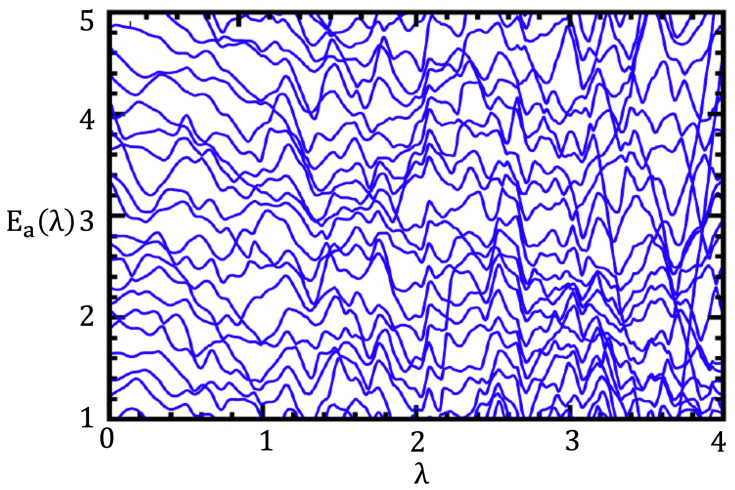
Levels of a model system as a function of the parameter λ. Each level may be visualized as a particle with the position given by the energy, $E_{a}\left(\lambda \right)$, the velocity given by the level slope $dE_{a}/d\lambda$, and acceleration (curvature of the level, $d^{2}E_{a}/d\lambda^{2}$.

**Figure 2 entropy-25-00491-f002:**
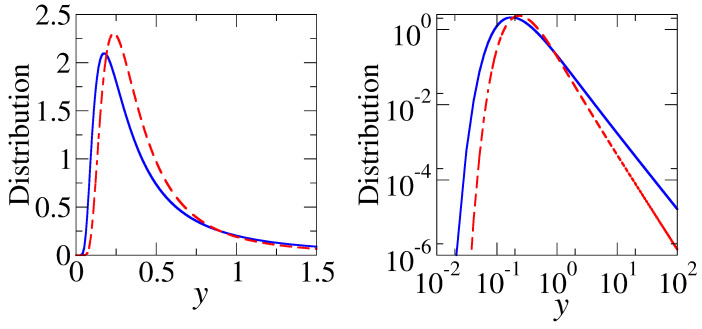
Scaled fidelity susceptibility distributions in the linear (**left**) and logarithmic (**right**) scales for GOE, Equation (33) (blue lines) and GUE, Equation (34). Observe a power law tail at large scaled fidelities, *y*.

## Data Availability

No original numerical or experimental data were produced for the present contribution.

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
