# Peer review of "Quantum Chaos and Level Dynamics"

_entropy, 2023, doi:10.3390/e25030491_

Round 1

Reviewer 1 Report

The paper focuses on a review about the level dynamics of quantum chaotic systems. With this purpose, the author discusses some relevant statistical mechanical adressings employed in the literature. The manuscript is well written and its reading can be accompannied without difficulties by both neofit and expert readers. However, some points (minor and major corrections) of the paper need to be corrected and/or improved in order to consider it for publication. The main concerns are listed below.

1) In line 5 of abstract "unversality" must be replaced by "universality". 

2) Rephrase the sentence beginning with "While the subject..." between lines 18 and 20 of introduction. 

3) In line 22 it must be "considers".

4) Delete "under consideration" of lines 23 and 24.  

5) In line 82 it must be "interpolating".

6) In line 83 "signifies" must be replaced by "means".

7) The paper lacks to mention an important, recent and geometric characterization of quantum chaos: Information Geometrodynamical Approach to Chaos (called briefly IGAC), where the information geometric entropy and the scalar curvature quantify dynamic transitions of chaotic systems described in terms of curved statistical manifolds,  see for instance [C. Cafaro, Mod. Phys. Lett. 22, No. 20, pp. 1879-1892 (2008)], [A. Giffin, S. A. Ali, C. Cafaro, Entropy 2013, 15(11), 4622-4633] and [I. S. Gomez, Physica A, 484, 117-131 (2017)]. My suggestion is to include this geometric approach in a new Section.  

8) It should be healthy for the completeness of the manuscript to include in some new Section the decoherence approach of the Hamiltonian modal interpretation [M. Castagnino, Gunzig, Int. J. Theor. Phys. 38, 47–91 (1999)],[M. Castagnino, O. Lombardi, Chaos, Solitons & Fractals 28, 879-898 (2006)],[M. Castagnino, Phys. Lett. A, 357, 97-100 (2006)], [I. S. Gomez, M. Losada, O. Lombardi, Entropy 19 (5), 205 (2017)], applied to clarify theoretical aspects between the classical limit and the chaotic emergence in quantum systems. This approach is located in an intermediate place between the stationary aspects (energy level characterization) and the dynamic aspects since it describes the weak limit as the representative state of the system (a coarse-grained state) in the asymptotic limit.

9) Conclusions must be improved including the remarks 7) and 8).

After addressing the above remarks the paper could be considered for publication. 

Reviewer 2 Report

The paper is a review of the level dynamics of the spectra of chaotic systems in dependence of some parameter. The author is one of the pioneers in the field, and hence best qualified to write such a report. The paper is clearly written, gives an informative overview suitable also for non-experts, and can be, as far as I am concerned, published with only minor changes. I start with some general remarks, before I am going to go through the paper step by step:

(i) The Pechukas-Yukawa model originally was introduced as a seemingly  promising path to explain the success of random matrix theory to describe the universal features of the spectra of chaotic systems.   Meanwhile this fact has been proven using periodic orbit theory, with the cap stone in the building by Fritz Haake and his group. Thus the original motivation to introduce the model has become obsolete. Of course the research in this field has a value of its own independent of the original motivation,  but a remark would be appropriate.

(ii) The paper concentrates on the  spectral level dynamics in closed systems, meaning that the eigenvalues may move exclusively on the real axis. In particular the sequence of eigenvalues can never change, whenever two of them approach, they are repelled form each other. A much richer phenomenology appears, if one allows for an eigenvalue level dynamics in the complex plane in dependence of, e.g., the coupling to external channels. I just would like to mention the phenomenon of resonance trapping. I do not ask the author to include a discussion of open systems, which clearly would exceed the intended scope of the review. But to mention of this field, and perhaps to cite some relevant papers  would be appropriate.

(iii) The experiments are mentioned in the very short paragraph 8. The message of this paragraph may be condensed in one single sentence:  "There are also experiments". This is a bit too little. I suggest to move this paragraph as a whole to the introduction, and in addition cite the relevant experimental results at the appropriate places in the text.

(iv)  There is not a single figure in the paper. For experts they are not needed, but if one has also non-expert readers in mind, one should show at least one figure in the introduction with a typical spectral level dynamics.  There are numerous examples in the literature.  Also in the following chapters the one or other figure might be appropriate: "A figure tells more than 1000 words!"

And now to the individual remarks. The numbers refer to the line numbers in the manuscript :

(60):  "L=1/2 Tr (F^2)":   The same quantity is called "Q" in Eq. (9). "L" probably refers to "angular momentum", see also line 71? I do not like this term in this context.  In one-dimensional systems there are no angular momenta.

Eq. (10):  One should avoid a wrap of the formulas if possible. There is sufficient place in one line.  Or is this done by the editor?  And the size of the brackets should decrease from the outside to the inside, not vice versa.  In the present version the formulas are difficult to read.  The same remarks hold for many of the successive equations.

Eqs. (11+14):  "exp" should be written in roman, not italics.

(70):  "...  integrated our ...":  Something is missing in the sentence.

(78):  "ensembe":  typo.  I suggest a spell-check of the manuscript.

(80): "alpha=1" is not motivated. Usually the constant is fixed by the demand to get a mean density of states of one, which would give here another constant.
"gamma/alpha=10^p" is odd. Why  the base "10" and not the canonical base "e"?

Eq. (14): The equation is a trivial special case of Eq. (11), no need to repeat it here!

(110): "localizaiton": typo.

Sect. 4: Gaussian velocity distributions are observed only for global perturbations, for local perturbations a K0-Bessel distribution is found [Barth et al, PRL 82, 2026 (1999)], which had been overlooked by Simons, Altschuler.  In a later paper this is corrected [Marchetti et al, PRE 68, 0336217 (2003)] . A short discussion of this point is appropriate.

(181): "hyndrogen": typo.

(193): Sect. 4 is on velocities, Sect. 5 on accelerations, Sect. 6 again on velocities. Odd! I suggest to interchange Sections 5 and 6.

Sect. 8: See point (iii) above.

(257-8):  "... a pure state ... where lambda ...":  Something is missing in the sentence.

Eq. (34): "I^{O,2}_N": What is the meaning of the upper index "O,2"?  It is not needed here. Either omit or explain it!

(278): Replace "proportional" by "asymptotically proportional"!

(295): "Tne", "sir": typos.
I know that Michael Berry does not like to be addressed as "Sir Michael", and in publications it definitely is common practice not to use the "Sir".

(313): The is no formula for the trace of G. So "given by formulae discussed above" probably refers to Eq. (39)?  Just out of curiosity: The trace is an invariant, and  this often simplifies calculations considerably, and sometimes allows get simple compact expressions. Is this the case?

(315): "samer": typo.

(325): "stady": typo.

(333): The abbreviation "MBL" is not introduced.

(335-7): Incomplete sentence.

(344): I am accustomed to use the term "breath-taking" in other contexts, but keep it if you like.

(359): "I am already anticipating the excitement":  Still curious and enthusiastic?! Keep it up!
